# Language models and protocol standardization guidelines for accelerating synthesis planning in heterogeneous catalysis

Manu Suvarna[1], Alain Claude Vaucher [2], Sharon Mitchell [1], Teodoro Laino [2] ✉ & Javier Pérez-Ramírez [1] ✉

Synthesis protocol exploration is paramount in catalyst discovery, yet keeping pace with rapid literature advances is increasingly time intensive. Automated synthesis protocol analysis is attractive for swiftly identifying opportunities and informing predictive models, however such applications in heterogeneous catalysis remain limited. In this proof-of-concept, we introduce a transformer model for this task, exemplified using single-atom heterogeneous catalysts (SACs), a rapidly expanding catalyst family. Our model adeptly converts SAC protocols into action sequences, and we use this output to facilitate statistical inference of their synthesis trends and applications, potentially expediting literature review and analysis. We demonstrate the model's adaptability across distinct heterogeneous catalyst families, underscoring its versatility. Finally, our study highlights a critical issue: the lack of standardization in reporting protocols hampers machine-reading capabilities. Embracing digital advances in catalysis demands a shift in data reporting norms, and to this end, we offer guidelines for writing protocols, significantly improving machine-readability. We release our model as an open-source web application, inviting a fresh approach to accelerate heterogeneous catalysis synthesis planning.

Heterogeneous catalysis stands at the forefront of developing sustainable technologies, driving the transition towards carbon-neutral chemicals and green energy carriers from renewable feedstocks[1,2]. At its core, catalyst design revolves around exploring and refining synthesis procedures, enabling the creation of unique and tailored architectures with distinct reactivity[3,4]. Catalyst design is often viewed as more of an art than a science due to the subtleties in preparation methods introduced by synthetic chemists, ultimately leading to diverse protocols and catalyst formulations found in research articles and patents[5-8]. Despite the empirical nature, the literature review serves as the foundational step in designing reaction-specific catalysts, providing crucial insights into synthesis strategies, suitable active sites, supports, and promoters, and preventing the repetition of past work[9,10]. However, with the ever-growing rate of publications, catalysis practitioners face a daunting task of keeping abreast of the latest developments in their respective fields. This challenge spans from identifying gaps for formulating original research objectives for projects to drafting grant applications and identifying opportunities for intellectual property in patents. Furthermore, traditional literature searches can be highly time-consuming, often spanning several weeks or months. This extended duration calls for more efficient methods for literature review and synthesis protocol extraction.

In recent years, text mining has gained prominence in automated information extraction from large corpus of material sciences[11-14] and

[1]Institute for Chemical and Bioengineering, Department of Chemistry and Applied Biosciences, ETH Zurich, Vladimir-Prelog-Weg 1, 8093 Zurich, Switzerland. [2]IBM Research Europe, Säumerstrasse 4, 8803 Rüschlikon, Switzerland. ✉e-mail: teo@zurich.ibm.com; jpr@chem.ethz.ch

organic chemistry[15–18] publications. Notable examples include named entity recognition to extract material properties,[19–22] and natural language processing[9,15,16,23–27] and deep learning[17,18,28,29] approaches to capture synthesis protocols and store this information in structured databases that enable collective insights. The emergence of large language models (LLMs), for example, GPT-3 and ChatGPT, claim to have a disruptive impact on natural sciences and engineering[30–32]. Generally, language models could read hundreds of synthetic procedures and (i) expedite the literature review and foster collective analysis of experimental data to identify interesting patterns and unexplored areas, (ii) generate data for training machine learning models to screen reaction-specific catalysts, and (iii) ultimately, drive computer-assisted synthesis planning and autonomous experiments to accelerate innovation in catalyst discovery and design[33,34]. Despite these promises, the application of text mining and language models in the heterogeneous catalysis community remains relatively unexplored.

In this paper, we introduce a transformer model for the automated extraction of synthesis protocols in heterogeneous catalysis, aiming to streamline the literature review and analysis process. The significance of the approach is illustrated by the case of single-atom heterogeneous catalysts (SACs), a catalyst family that has garnered substantial attention in recent years due to their precise atomic-scale structures, high metal utilization, and unique reactivity[35–37]. A literature comparison reveals that SACs are the fastest-growing family of catalytic materials (Fig. 1a) over the past decade, with a wide range of electro-, thermo-, and photocatalytic applications (Fig. 1c). However,

their compositional diversity combined with challenges associated with confirming their properties and the fast-paced exploration of synthetic routes, including wet-chemical, solid-state, gas-phase, and hybrid methods[38–40] or minor adaptations to existing procedures[41–43] (Fig. 1b), make it extremely challenging to follow progress.

Based on a transformer architecture, our model captures details contained within and converts prose descriptions into action sequences with associated parameters, covering all steps required for replicating the synthesis (Fig. 2). We extend its application to analyze trends in prominent electrocatalytic processes like oxygen and carbon dioxide reduction reactions. Beyond SACs, our model is adaptable, offering accurate predictions for other prominent families of catalytic materials. Its potential to reduce literature analysis times significantly underscores its value. However, our analysis highlights the critical issue of non-standardized synthesis reporting on text mining efficiency. To address this, we propose guidelines for machine-readable synthesis procedures. By comparing the model on original and guideline-modified protocols, we observe a significant performance enhancement, demonstrating the value of protocol standardization as a key enabler to accelerate synthesis planning in heterogeneous catalysis.

## Results

### Annotation and extraction of synthesis actions

In our study, we targeted reviewing SAC literature from when the term was introduced in 2010 to 2021, identifying nearly 1200 papers. Of

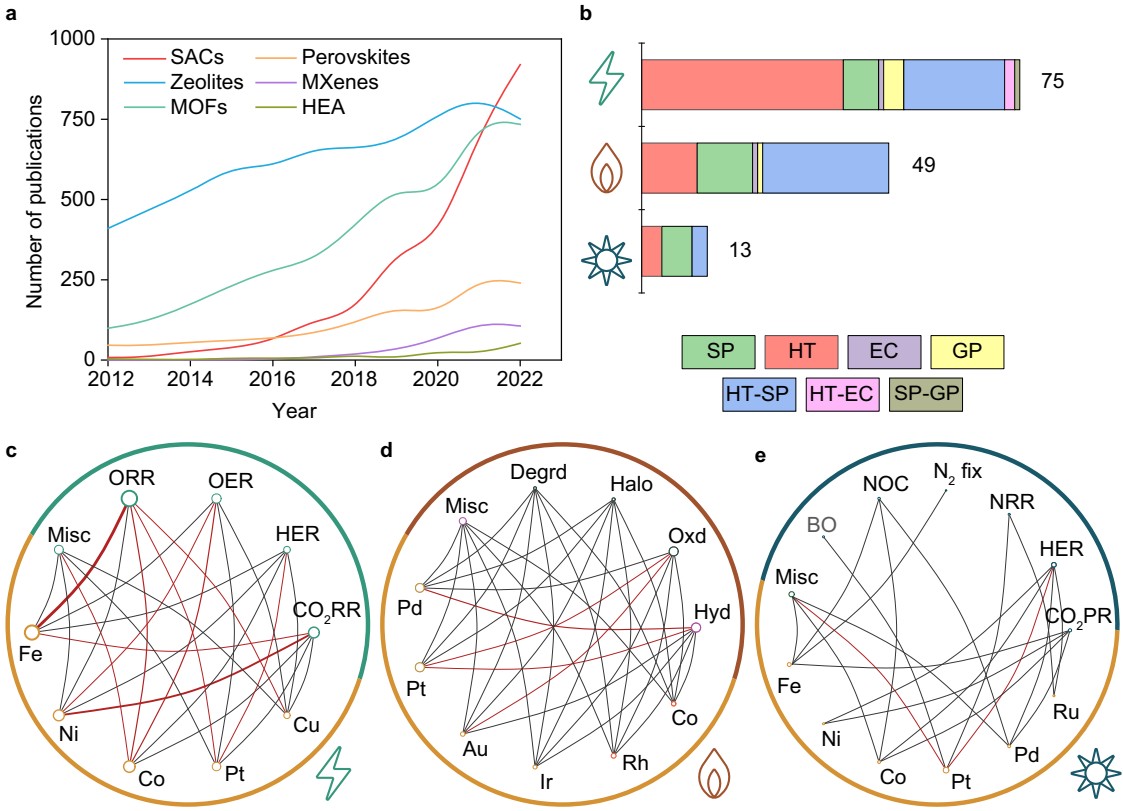

**Fig. 1 | Trends in SAC research. a** The Comparative increase in the number of publications on SACs to other prominent material families for catalytic application illustrates the exceptional growth. **b** Relative number of publications reporting SACs for electro, thermo, and photocatalytic applications based on manual analysis, illustrating the distribution of synthetic approaches applied. **c–e** Network maps linking the main metals supported in SACs to their targeted electro-, thermo- and photocatalytic applications. Node areas are scaled to the frequencies mentioned in literature, and the edges interconnecting the nodes relate to the frequency with which the SACs and reactions appear simultaneously. The nodes' size and the edges' thickness are normalized to enable a fair comparison across all three catalytic applications. MOFs metal-organic frameworks, HEA high entropy alloys, SP solution-phase, HT high-temperature, EC electrochemical, GP gas-phase, and their corresponding hybrid methods; ORR oxygen reduction reaction, OER oxygen evolution reaction, HER hydrogen evolution reaction, CO$_2$RR carbon dioxide reduction reaction, Halo halogenation, Hyd hydrogenation, Oxd oxidation, Degrd degradation, NRR nitrogen reduction reaction, NOC nitric oxide conversion, and BO benzaldehyde oxidation. Source data are provided in the source data file.

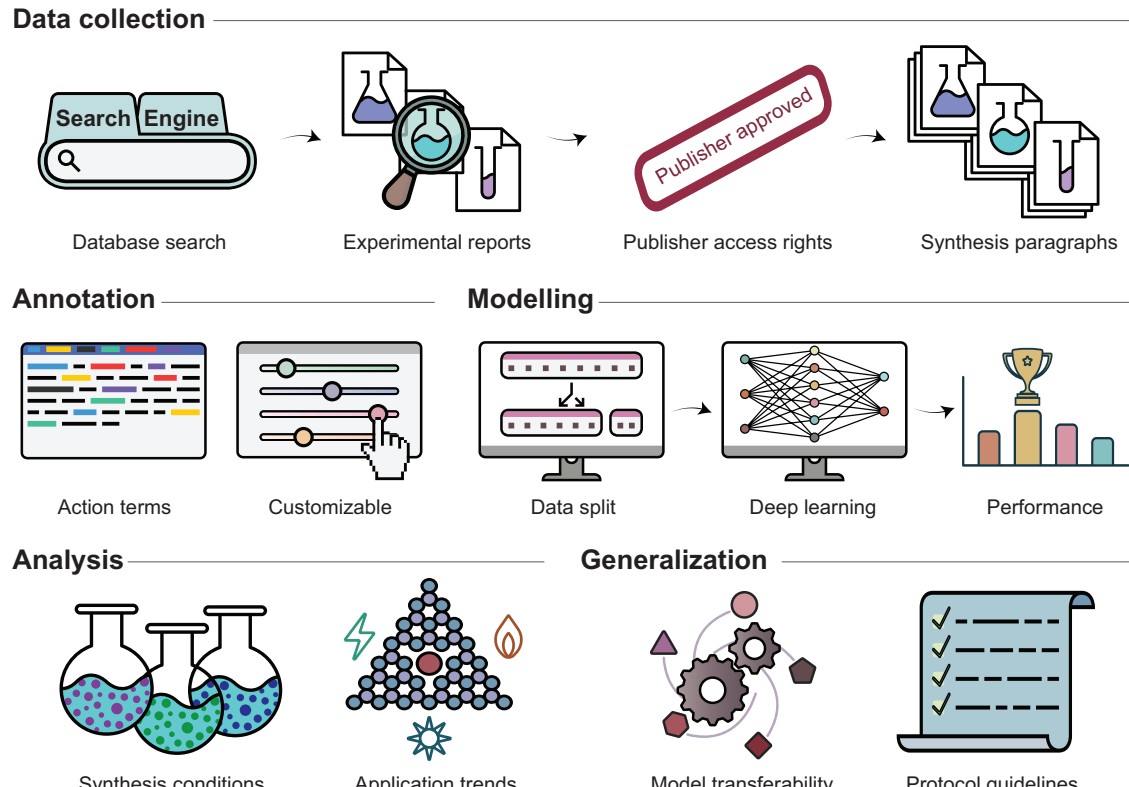

**Fig. 2 | Approach to text mining SAC literature.** A literature search identified articles on SACs published between 2010-2021, followed by manual classification of experimental reports. Relevant papers were sourced with permission from various publishers in json or txt format. To configure the annotation framework, 127 synthesis paragraphs containing 936 sentences were manually annotated with 33 action terms defined in this study. Post annotation, the 936 sentences were split into training, validation, and test split (80:10:10), and the ACE model was developed until achieving acceptable performance. Analysis of the data extracted by the model provides insights into synthesis and application trends. Extension of the model to other families of heterogeneous catalysts demonstrates its transferability, and guidelines for the standardization of synthesis protocols set the path for facilitating future text mining endeavors.

these, 569 were experimental reports, while the remainder comprised reviews and theoretical studies. We focused on experimental papers, from which we manually reviewed 145 publications, aiming to analyze the trends in synthetic routes investigated across various thermo-, electro-, and photochemical reactions (Supplementary Note 1). Synthesis routes of SACs typically encompass various steps, such as mixing, wet deposition, pyrolysis, filtering, washing, and annealing. Conventionally, these procedures are reported within the "Methods" sections of scientific articles as unstructured natural language-based textual descriptions. Our goal was to extract all relevant synthesis steps, related conditions, and the resulting material compositions into a structured format, as exemplified in Fig. 3a. To facilitate our analysis, we defined the synthesis of SACs as the process starting from the metal precursor and carrier or carrier precursor and continuing until a catalytic material containing single atoms stabilized on a support is obtained. Using this definition, we classified the reported synthetic approaches into eight categories and an additional post-synthetic treatment step (Supplementary Table 1).

To develop a framework for extracting and analyzing this information, we initially identify the most commonly used synthetic steps that could later be used as action terms for annotation purposes (Supplementary Note 2, Supplementary Table 2). We further analyzed the frequency of occurrence of synthetic steps in the same subset of 145 publications through manual analysis. Each step involves several relevant parameters, such as the temperature, temperature ramp, atmosphere, and duration in the case of pyrolysis. These details need to be clearly defined and can be customized depending on the required level of detail. Thus, all such synthetic steps and essential parameters necessary to replicate the experiments were identified and labeled as action terms (Supplementary Table 3).

The action terms were then used to manually annotate a randomized subset of 127 synthesis paragraphs (approximately 25% of the total paragraphs compiled) comprising 936 sentences. The annotation was performed on the dedicated software, as shown in Fig. 3b. (see "Methods" for details). Using this set of annotated paragraphs, and combining it with previously annotated set for organic synthesis,[17] we fine-tuned a pretrained transformer model[17] to devise our ACE (sAC transformEr) model, which translates full-length unstructured sentences from entire paragraphs into a structured, machine-readable sequence of information (see "Methods" for details and Supplementary Note 3). We evaluated the fidelity of our model based on metrics such as the Levenshtein similarity and BLEU (Bilingual Evaluation Understudy) scores. With an overall Levenshtein similarity of 0.66, our ACE model can capture and extract approximately 66% of information from synthesis protocols into correct action sequences, while a BLEU score of 52 attests to the high-quality translation of synthesis sentences from natural language into machine-readable formats (Supplementary Note 4)

To maximize the potential of this language model for automated extraction of synthetic protocols, we wanted to make it easily accessible to a broad audience and, more specifically, experimental researchers. For this purpose, we developed a web application and open-sourced it at synthesis_protocol_extraction. Synthetic and computational researchers can easily access this tool, requiring no prior programming or coding experience.

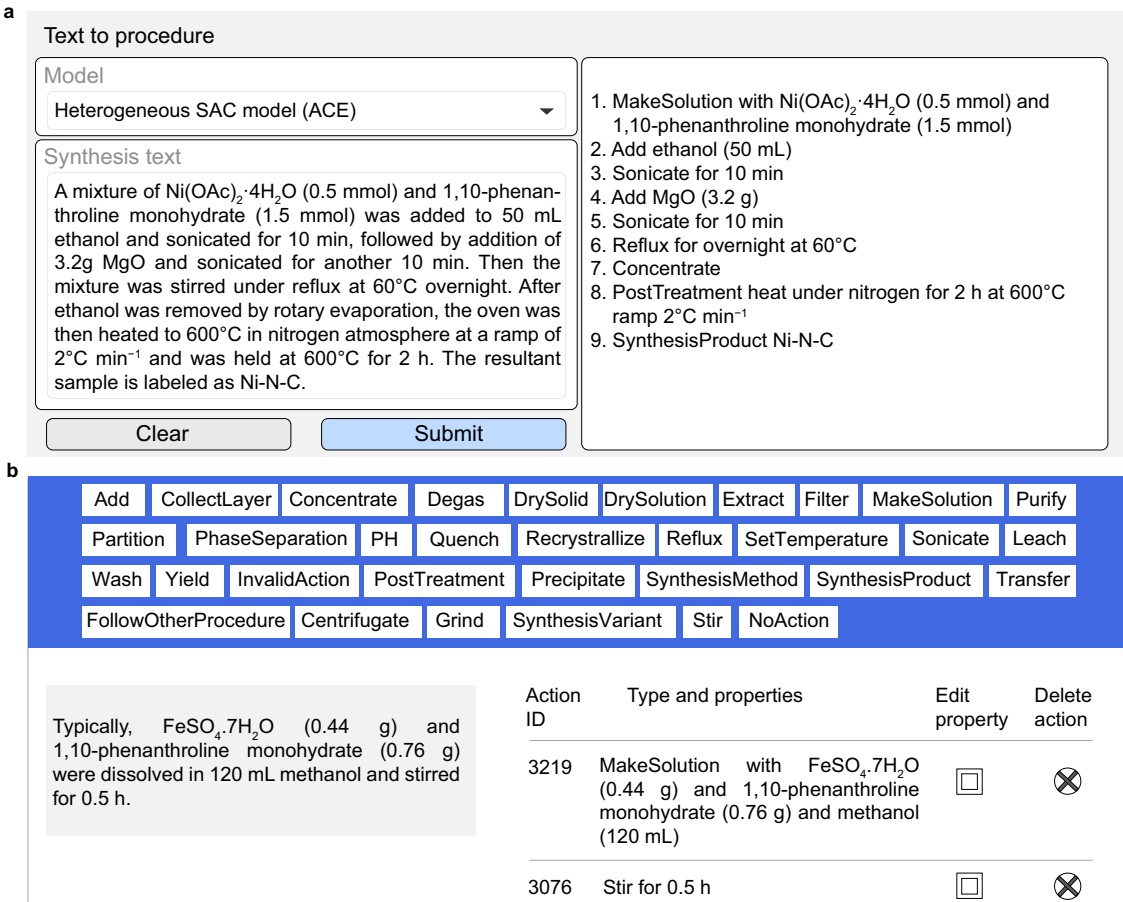

**Fig. 3 | User interface of the machine-reading platform. a** Software interface illustrating the model capabilities, users input synthesis paragraphs in the left panel, and the model output in sequential actions are displayed in the right panel. **b** The annotation tool for adding and editing action items corresponding to a synthesis sentence, displaying the complete list of action terms (blue panel). The sentence to annotate is displayed on the left, while the annotation procedure is depicted on the right. Selection of the edit property button opens a separate tab where the action can be edited.

## Accelerating literature analysis and synthesis insights

The ACE model serves the dual purpose of synthetic protocol text mining and collective analysis of this literature data to reveal interesting patterns. As such, this model serves as an invaluable tool to catalysis practitioners as it expedites literature review practice, which is often routine, tedious, and time-intensive. We estimate the time spent on one single paper and for collecting details on the metal speciation, composition, synthetic route, and reaction types by a SAC researcher to be approximately 30 min without any help and under 1 min with our ACE model. Scaling this effort to 1000 publications would cumulatively result in a minimum of 500 man-hours in an ideal case scenario, while text mining these publications by the ACE model would take a mere 6–8 h (a set of 300 papers were text mined in less than 2 h for quantification) offering over 50-fold reduction in the time invested for literature analysis- and thus demonstrating the true worth of language models.

The results of our analysis on the synthesis of SACs complement existing domain knowledge, thereby instilling greater confidence in the model output. By applying topic queries to articles reporting SACs for the oxygen reduction reaction (ORR) and $CO_2$ reduction reaction ($CO_2RR$), applications accounting for approximately one-third of the reports in our database, we identified the most frequently used metals and metal precursors, carrier materials, and solvents (Fig. 4a, b). Our analysis revealed that Fe is one of the most commonly investigated metals for the ORR reaction, with Fe-based precursors typically involving chlorides or nitrates. The model findings also revealed that carbons derived from zeolitic imidazolate frameworks (ZIF-8) are a popular choice for carrier materials in ORR applications due to their high surface areas, microporous structures, chemical and thermal stabilities, and synthesis controllability, with ZIF-8 synthesized from 2-methylimidazole and $Zn(NO_3)_2$ via solvothermal and solvent methods being prevalent. The analysis also provides valuable insights into the temperatures applied during thermal treatments in SAC synthesis, for example, pyrolysis, annealing, and reductive treatments (Fig. 4c). A broad range of temperatures are used in all cases, but distinct peaks are observed. These are typically around 1173 K for annealing and pyrolysis steps. Reduction treatments usually activate the catalyst at lower temperatures (373–423 K). For SACs subjected to heat treatment, two distinct peaks are observed, where the evidenced distribution at lower temperatures could plausibly account for instances where the catalyst was subjected to heat treatment (573–623 K), post the metal stabilization on the carrier.

Though the above enterprise presents valuable information to synthesis experts for informed decision-making during the experimental planning stages, we realize that the seamless automation of text-mined synthetic protocols to create comprehensive SAC repositories or databases is far from realization and requires extensive manual intervention. We highlight and attribute this limitation to the absence of uniform data presentation and lack of standards in writing protocols[6,44,45]. This discrepancy consumes valuable time and introduces the risk of misinterpretation and inaccuracies in the process. For example, we observe that the commonly used solvent ethanol is referred to using various terminologies such as 'ethanol,' '$C_2H_5OH$,' 'EtOH,' and 'EOH' in different publications. While human researchers can recognize these variations as referring to the same chemical entity,

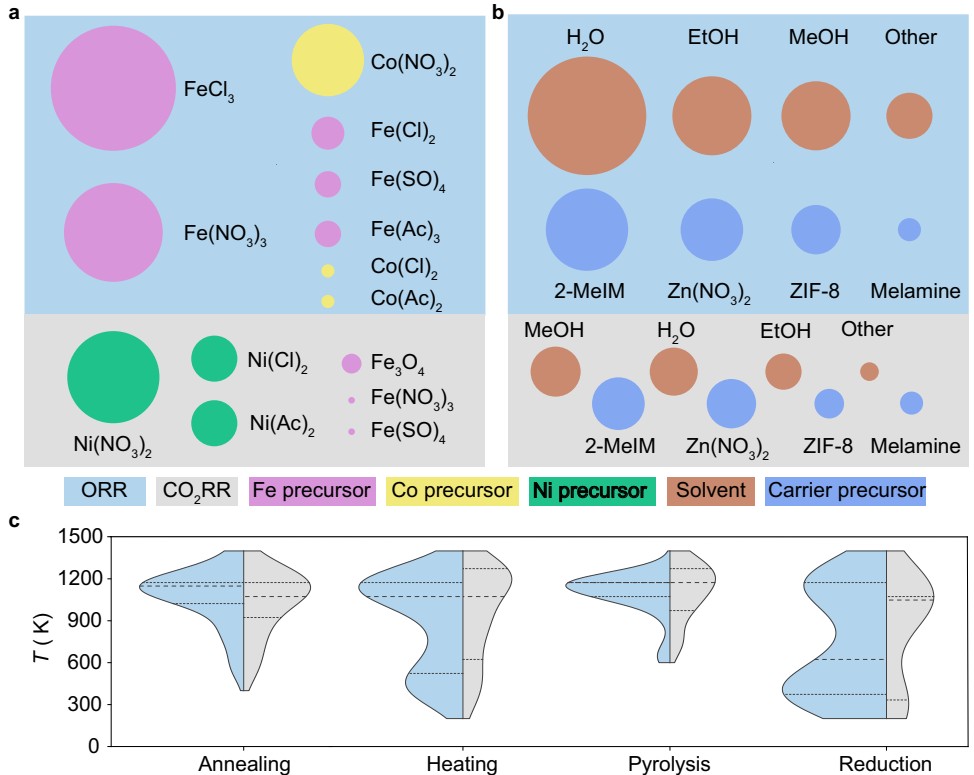

**Fig. 4 | ACE-derived synthesis landscape of SAC literature.** Overview of the most frequently reported **a** metal precursors and **b** carriers and solvents in SACs developed for ORR (blue rectangles) or CO₂RR (gray rectangles) identified by machine-reading of the complete dataset. The areas of each sub-rectangle are scaled to the frequency count of the respective reactions, while the circle diameter corresponds to the adjusted normalized count of the chemical species. EtOH ethanol, MeOH methanol, 2-MeIM 2-methylimidazole, and ZIF-8 zeolitic imidazolate framework-8. **c** Plots summarizing the temperatures reported for distinct thermal treatments applied in synthesizing SACs for ORR and CO₂RR. Source data are provided in the source data file.

language models often treat them as distinct substances. This issue becomes increasingly complex when dealing with multiple such precursors, solvents, and associated chemicals presented under various nomenclatures. Similarly, we notice that the term 'heating' is synonymously used for pyrolysis, carbonization, or annealing, thereby rendering challenges for the model to determine the appropriate thermal-treatment method. Developing a language model capable of accurately identifying and accounting for all such chemical entities and unit operations is formidable. In light of these challenges, we emphasize the critical importance of data standardization in scientific publications. Only through enhanced data uniformity can we pave the way for broader applications of language models in chemistry and catalysis-related tasks, thereby advancing interdisciplinary research and innovation in these fields.

**Model adaptability and limitations**

Visualizing the data extracted by the model sheds valuable insights into the unit operations related to synthesis across publications. Our findings reveal a close alignment between the distribution of action terms predicted by the ACE model and the annotated synthesis steps, also referred to as ground truth (Fig. 5a). Notably, the ACE model demonstrates a remarkable accuracy in predicting action terms such as ThermalTreatment, SynthesisProduct, Centrifuge, Transfer, etc. Furthermore, we note the recurrent presence of action terms such as Add, Stir, MakeSolution, ThermalTreatment, DrySolid, Wash, and Concentrate, etc., in the distribution list. This observation aligns with our initial manual analysis, where we statistically inferred the predominance of solution-phase and high-temperature procedures and their hybrid forms in SAC synthesis, and these encompass the aforementioned action terms. This, in turn, demonstrates the model's prediction prowess.

Here, we rationalize that since the ACE model is purely data-driven and effectively identifies action steps related to solution-phase and high-temperature synthesis methods, its predictive capabilities should not be limited to SAC protocols. Instead, it has the potential to be adapted for a broader category of heterogeneous catalysts that are prepared by similar synthetic routes. Within heterogeneous catalyst research, supported mono-, bimetallic, and promoted systems are amongst the most investigated materials, often prepared using wet-chemistry techniques like precipitation, impregnation, and sol-gel methods, among others. Extending the model's application to extract synthetic protocols of these catalysts would prove its ability to generalize across various catalyst types.

To test this hypothesis, we selected a randomized subset of 18 synthesis paragraphs from the literature on a few prominent reactions, including CO₂ hydrogenation, acetylene hydrochlorination, and Fischer-Tropsch synthesis. These paragraphs were subjected to the ACE model prediction, resulting in 202 action sequences (Supplementary Note 6). To determine the model accuracy, we devised a simple yet effective evaluation metric, termed human machine-readability index (HMI) (see Methods for details). Here, we observed that of the total sample size, the ACE model could predict 125 actions correctly, inclusive of all essential information, 28 sentences were partially correct, while 49 sentences were predicted wrongly, resulting in the HMI of 69%. (Fig. 5b) To put into context, this inferred that the model could extract approximately 69% of information from the evaluated paragraphs, implying that it could generalize and extract synthesis details with accuracies similar to SAC synthesis as described in the earlier section. These findings quantitatively prove that the ACE model can be extended to extract operations from texts of other families of heterogeneous catalysts with minimal training data or the

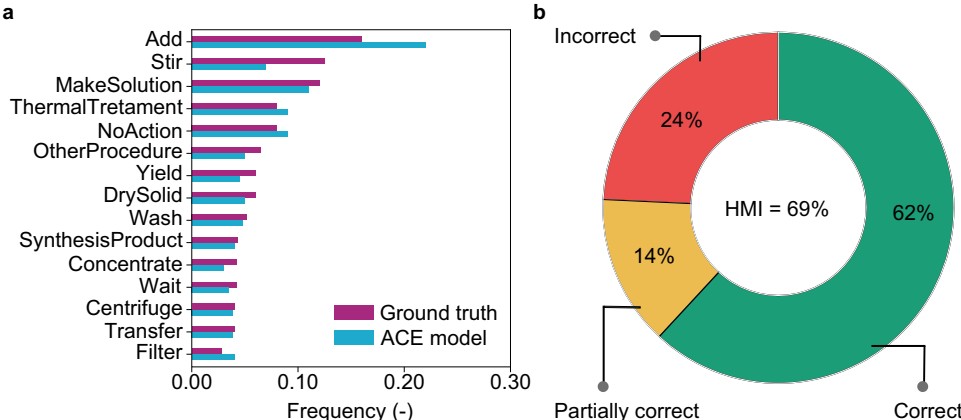

**Fig. 5 | Predictive and generalization capabilities of the ACE model.**
**a** Distribution of action types for the test set, determined by ground truth (human annotation) and by the ACE model. The action terms are ordered in decreasing frequency of the ground truth. **b** Generalization of the model to a broader class of heterogeneous catalysts. A sample of 18 paragraphs encompassing synthetic protocols on mono-, bimetallic, and promoted catalysts with 202 action sequences resulted in an HMI of 69%. Source data are provided in the source data file.

definition of new action types to cover different catalytic reactions of interest.

However, it should be noted that the automated extraction of action sequences for SAC synthesis is a first step in this direction, an ongoing effort with its limitations, including the current accuracy level, inability to extract details of grouped information, and failure to capture gas and vapor phase and electrochemical routes (Supplementary Note 7). For instance, when the model encounters a synthetic protocol where the catalysts are prepared with different metal contents or activated at different temperatures, where all the experimental details are written in parenthesis, the model typically retrieves one of the given values or assigns an arbitrary value to the respective action term. Furthermore, the predictive prowess of the model is inherently compromised on encountering less standard synthesis protocols, for example, gas-phase or electrochemical. Based on the model prediction, it is evident that essential procedures like plasma treatment and configuration of the plasma chamber for gas-phase and details including type of electrodes, calibrations, and cycles for electro are currently not well captured. The poor generalization is primarily due to the absence of these steps as action terms due to the scarcity of these methods in the training set of paragraphs for annotation. This is a significant limitation to bear in mind when using the model for prediction or extraction in these areas. Nevertheless, we believe that curating action terms specific to these routes and supplementing the model training with these potential new terms will scale its accuracy proportionally.

## Improving machine-readability through protocol standardization

Examination of the incorrect predictions of the ACE model reveals that the errors are often minor, involving the omission of one or two action terms. These errors are frequently acceptable alternatives to the ground truth. To improve the model performance, we propose two systematic approaches. First, we recommend increased annotation of synthesis paragraphs followed by model retraining and the addition of new annotation terms corresponding to synthesis steps where the model is least confident. Second, we advocate for standardizing the way synthesis protocols and experimental results are reported in literature to facilitate efficient text mining and automated data analysis. Towards this goal, we offer eight guidelines to standardize synthesis paragraph writing (Fig. 6a, b).

(i)  Publishers and journals should require detailed synthesis protocols as Supplementary Information to free authors from word limits and (self-)plagiarism constraints, allowing for more accurate descriptions, which would improve synthetic reproducibility and model training. (ii) Chemical composition and targeted properties (e.g., metal nuclearity in supported metal catalysts) of synthesized catalysts the procedure used to prepare them should be explicitly mentioned to ensure accurate correlation of synthesis details with samples.

(ii)  Supplier details and grade of all chemicals used in the study should be given in a uniform format. For our model, providing them as a single sentence at the beginning of the paragraph rather than listing them at the first occurrence of the chemical entity improves the performance.

(iii)  Specify exact quantities of chemicals rather than writing mathematical expressions in ratios will improve machine readability, which would otherwise require the development of specific calculations to decode the formulation, entailing separate challenges.

(iv)  Keep protocols concise, excluding elaborated discussion of choices of synthetic steps. Additionally, separate descriptions of characterization techniques from synthesis paragraphs.

(v)  Provide details on all samples reported rather than focusing on the best-performing catalyst. If multiple samples, for example of distinct composition, are prepared following the same synthetic procedure, we propose that such details and the speciation of these SACs be separately listed in a tabular format after the synthesis paragraphs.

(vi)  Use standardized terms for synthesis steps and catalyst vocabulary. This requires more effort from the scientific community to develop and adopt standardized definitions.

(vii)  Publish all synthesis-related datasets with every manuscript for transparency and reproducibility.

We performed a demonstration test to evaluate our premise that protocol standardization can improve machine-readability (Supplementary Note 8). From the pool of SAC literature compiled in this study, we randomly selected 11 paragraphs, where the SACs were prepared by wet-chemistry routes and labeled them as pristine paragraphs. Subjecting these paragraphs to the ACE model resulted in 151 action sequences, where we observed fully correct and partially correct predictions on 99 and 19 actions, respectively, while 33 were predicted incorrectly, resulting in an HMI of 72%. We then modified the pristine paragraphs to comply with the first five proposed guidelines, which are straightforward to implement, and termed them as standardized paragraphs. On assessing the model on these paragraphs, 156 actions sequences were generated, comparable to the pristine action set. Interestingly, here, the correct

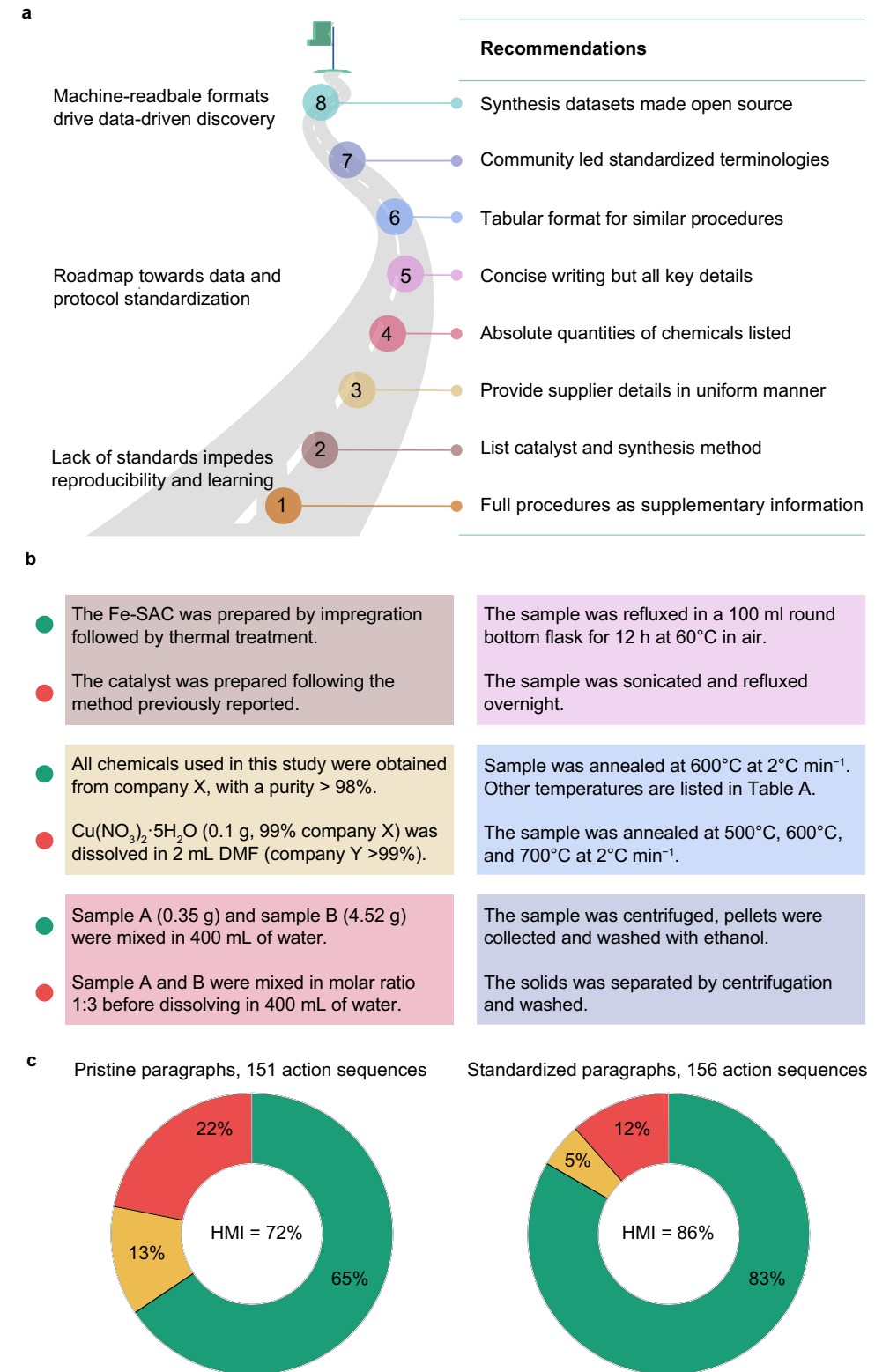

**Fig. 6 | Standardization of synthesis protocols for enhanced text mining. a** A set of eight guidelines are identified and recommended as the outcome of this study to assist text mining endeavors in catalysis. **b** Specific examples outlining best practices in writing synthetic protocols that comply with the recommended guidelines. The green and red circles to the left of each example represent compliance and non-compliance to the guidelines **c** Model performance using the human machine-readability index (HMI) on 11 paragraphs before and after standardization. The color codes used include red, yellow, and green for incorrect, partially correct, and correct predictions, respectively. Source data are provided in the source data file.

action sequences increased to 130, while the partially correct and incorrect actions were reduced to 8 and 18, respectively. This, in turn, resulted in an HMI of 86% - a significant 14% increase compared to the pristine counterpart (Fig. 6c). It is worth mentioning, in comparison to increasing training set size or synthetic data augmentation as discussed in the earlier section, where we notice meager increments in model performance (c.a. 2–3%) under various scenarios, data standardization offers a more promising alternative to improve model fidelity and machine-readability. More importantly, as a proof-of-concept, we successfully demonstrate the significance of protocol standardization and its indispensable role in machine-reading and NLP-related tasks in chemistry and catalysis.

The guidelines proposed for protocol standardization are not limited to the above cause but can be conveniently translated to manage data in heterogeneous catalysis as per the FAIR (Findable, Accessible, Interoperable, and Reusable) principles[44,46]. For instance, in addition to the synthesis protocols being written in the Supplementary Information, authors can also submit the same in online repositories for easier and enhanced accessibility. Along with the metadata of the protocols including, but not limited to who, when, and where the synthesis was performed, details on the supplier and grade of the chemicals and equipment or reactors used during synthesis will allow for greater reproducibility of the protocols complying with FAIR principles. An alternative to online repositories would be to list the details of the synthesis protocols, including catalyst properties and synthesis approach, quantities of precursors, solvents, and supports, and the respective unit operations in a succinct manner in Excel sheets or customized electronic lab notebooks (ELNs) in tabular formats, such that all the above information are efficiently managed through the life-cycle of the project. These synthesis datasets should be published with every manuscript. Furthermore, details of similar procedures could be seamlessly listed in tabular formats within ELNs, making data interoperable. Here we highlight that the objective of protocol standardization by no means intends to limit the creativity of the experimental researcher, but rather encourage scientific writing and data reporting practices, so that it can be translated to meet the requirements of text mining approaches. Another forward looking alternative would be to write synthesis procedures as a sequence of actions rather than plain texts. We believe, such practices if fostered, would encourage experimentalists to list procedures in the correct sequence of actions, along with all necessary details essential to replicate the experiment. Such formats would not only benefit the experimental community, but also be greatly advantageous for text mining and language models endeavors. Overall, through such protocol and data reporting standardization efforts, which inadvertently improves the quality of data, we anticipate text mining approaches, be it natural language processing, association rule learning, sequential pattern mining, pattern tracking, etc., to effectively work with small datasets and minimal training, ultimately leading to their greater acceptance is chemistry and catalysis research.

## Discussion

Our study demonstrates the tremendous potential of transformer models to extract synthesis actions from heterogeneously catalyzed experimental procedures. These tools pave the way for data-driven design and guided synthesis of application-specific heterogeneous catalysts. Importantly, this methodology is not restricted to a specific class of catalyst, as demonstrated through various examples. While text mining and automated data collection in catalysis offer immense potential for driving the concepts of autonomous and robotic experiments, existing limitations arising from the lack of standardization in reporting synthetic protocols must be addressed. Based on the outcome of this study, we believe that current literature may not be sufficiently ready to capitalize on all the benefits of language models and machine learning tools.

For the catalysis community to fully benefit from the rapidly evolving data-driven technologies, significant efforts must be directed toward data collection, curation, and reporting practices. While these endeavors can be initiated with individual research groups, their global awareness and adoption will be only possible through community efforts led by national and international initiatives, including research data management and standardization. Furthermore, as language models become increasingly prominent, text mining in catalysis should be a collaborative endeavor involving researchers developing catalytic materials and experts in deep learning algorithms and text mining pipelines. This partnership will drive the emergence of novel concepts for digitized experiments. With improvements in language models and developments that enable reporting synthesis protocols in standardized and machine-readable formats, we envision the rapid growth of automated synthesis and data-driven discovery of catalysts, promising an exciting new era of discoveries in catalysis and chemistry. As a follow-up to this study, we aim to make the ACE model generative in nature. To accomplish this goal, we plan to standardize published paragraphs based on the guidelines provided in this study via prompt engineering in GPT-4, followed by retraining of our ACE model on the modified paragraphs.

## Methods

### Article collection

A comprehensive screening of SAC literature was performed in Web of Science and Scopus for the data collection process. This was done through keyword search, including terms such as ("single-atom catalyst" AND "electrocatalytic OR "thermocatalytic" OR "photocatalytic" OR "organic reactions") and ("SAC" AND "electrocatalytic OR "thermocatalytic" OR "photocatalytic" OR "organic reactions") during the period 2010–2021. The keyword search returned over 1200 works, including experimental, computational, and review articles on the topic. Since the main essence was to develop a text mining protocol specific to the synthesis procedures, the purely computational and review papers were discarded. At the same time, experimental works were retained, bringing the effective collection to 569 articles. The major publishers, including American Chemical Society (ACS), Royal Society of Chemistry (RSC), Elsevier, Springer, and Wiley, accounted for 97% of the publications. With support from the ETH library, appropriate agreements were signed with all the publishers to ensure freedom of text and data mining on these published articles and open access to the data extracted from the reports. The papers were procured in electronic formats such as json, html, or txt.

### Statistical analysis of synthesis methods

Based on the data compiled, a descriptive analysis to quantify various synthesis methods and the application of SAC across thermo-, electro-, or photocatalytic applications was performed on randomly selected 145 publications. The diverse synthesis methods were categorized into eight classes with an additional post-treatment step based on literature description and domain expertise (Supplementary Table 1).

### Annotation

Annotation is the process of assigning meaning to natural language text, which can apply to short phrases, long sentences, or entire paragraphs. It provides the AI model with essential information to learn the meaning or context of the text[11]. Text annotation parallels the data labeling procedure in classical machine learning, as this information is used to train the model and accurately interpret the text under consideration. Herein, 33 action terms were utilized for annotation, 27 of which were identified based on the most frequent action items in wet-chemistry labs,[17] and 6 catered to synthesis protocols written in the heterogeneous catalysis community (Supplementary Tables 2, 3). Once finalized, each sentence in the synthesis paragraph was manually annotated using these action terms on the modular

annotation framework based on the open-source tool doccano (Supplementary Table 4). Details on the working philosophy of this tool are described elsewhere[17,47]. Briefly, doccano offers a user-friendly web interface, which allows the annotator/user to add, edit, reorder, and confirm action steps on each of the sentences from the experimental procedures that appear on the front screen. This tool also has a customizable backend, offering greater flexibility for developers to add, edit, and delete action terms relevant to the project scope. In this study, 127 paragraphs, including 936 sentences, were manually annotated, and an annotation guideline was developed to ensure consistency throughout the study.

### Transformer model

The ACE model developed for translating experimental protocols to action steps was based on the transformer architecture[48]. The algorithm relies on the transformer encoder-decoder architecture, which creates a latent vectorized representation of texts, and its 'attention' functionality facilitates recognition of the essential and core parts of a sequence necessary for comprehending the textual meaning. The ACE model was developed by transfer-learning from a previously reported model,[17] by fine-tuning the pretrained model on a set of 936 annotated sentences compiled from 127 SAC synthesis paragraphs and 2295 annotated sentences from the realm of organic chemistry[17].

The ACE model is implemented using the OpenNMT-py library[49] with specific architecture, implementation, and hyperparameters (Supplementary Note 3, Supplementary Tables 5, 6). The entire dataset of 3231 annotated sentences was split into 80:10:10 ratios and labeled as training, validation, and test sets, respectively. The training-validation-test split was carried out in a randomized manner at the sentence level through the following steps. First, the number of sentences assigned to the train, validation, and test sets were determined based on the desired sizes (as a percentage of the total). Second, the full set of sentences and associated annotations were shuffled, randomizing their order. Third, the shuffled dataset was split into three contiguous subsets of the desired sizes for the respective splits. Furthermore, the model was progressively evaluated on 25%, 50%, 75%, and 100% of the total annotated data to evaluate the effect of data size on performance. The effect of data augmentation was also examined, where the size of the above-annotated sets was synthetically increased by 10-fold. (Supplementary Note 4). For reference, a 10-fold augmentation on 100% of the training data corresponds to 25820 synthetically generated sentences with annotation[17]. The model was trained for 20,000 steps in all cases, and checkpoints were saved every 1000 steps. The model checkpoint leading to the highest accuracy in the validation set was selected for analysis.

### Performance metrics

This study used various performance metrics, including the BLEU (Bilingual Evaluation Understudy), Levenshtein score, and the human machine-readability index (HMI). The BLEU score is a metric for evaluating machine-translated text, where values of 40-50 infer high-quality translations. At the same time, those below 25 imply poor translation and are not recommended[48,50] The Levenshtein score calculates the similarity index between the synthesis procedure in the actual paragraph and the action sequence predicted by the model[17]. Thus, the 100% accuracy refers to the fraction of sentences wherein the entire action sequence was predicted correctly, along with the respective properties. Though these metrics are conventional for NLP tasks, their inherent limitation is the need for annotation on the selected paragraphs under investigation, as generating such annotated datasets is usually time- and resource-intensive and may not be feasible in all cases. Moreover, the various synthesis-related parameters extracted by the model, necessitate validation, best performed through domain knowledge. However, currently, this scope is limited by the aforementioned metrics.

To address these challenges, we developed an effective metric, HMI, which brings forth the human factor in evaluating these model predictions. Briefly, a human expert manually checks the synthesis paragraphs and the corresponding sequence actions predicted by the ACE model and grades the output depending on its accuracy as described in Eq. 1.

$$HMI = \frac{I(0) + PC(0.5) + C(1)}{A_T} * 100 \qquad (1)$$

where, I = incorrect actions PC = partially correct actions C = correct actions $A_T$ = total actions

The model is graded as per the HMI score in the following manner:

- If the model predicts the correct action terms and extracts the corresponding material properties or unit operation values, it gets a full score of 1.
- If the model predicts the correct action term, but extracts incorrect material properties or unit operation values, then it gets a score of 0.5.
- If the model predicts an action term incorrectly, it gets a score of 0.

For example, if the output of a typical SAC synthesis paragraph results in a sequence of 10 action terms with five correct, two partially correct, and three incorrect predictions, this results in an HMI of 60%.

## Data availability

The curated dataset compiled in this study is open-sourced and deposited at Zenodo (https://doi.org/10.5281/zenodo.10033139). We released an open-source web application of our ACE model at synthesis_protocol_extraction. Further data supporting the findings of this study are available in the Supplementary Information. Source data for all figures in the manuscript and Supplementary Information are provided with this paper. Source data are provided with this paper.

## Code availability

A Python library with the definition of the new actions, the pre-processing of the data, and fine-tuning of the transformer model can be found on GitHub at https://github.com/rxn4chemistry/sac-action-extraction.

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

## Acknowledgements

This study was created as part of NCCR Catalysis (grant number 180544), a National Centre of Competence in Research funded by the Swiss National Science Foundation. We thank C. Ko, D. Faust Akl, S. Jaydev, Dr. G. Giannakakis, and Dr. A.J. Martín for the fruitful discussions on the manuscript and help with illustrations.

## Author contributions

M.S., T.L., and J.P.R. conceived and designed the project. M.S. led the data collection, annotation, model analysis, and manuscript preparation efforts. A.C.V configured the annotation platform and trained the transformer model. S.M. contributed to the statistical analysis and classification of synthetic methods. T.L. supervised the work. J.P.R. supervised the entire

project and managed resources and funding. All the authors provided inputs to the manuscript and approved the final version.

## Competing interests

The authors declare no competing interests.
