## [Peer review file · Nature Communications]

Reviewers' comments:

Reviewer #1 (Remarks to the Author):

The manuscript by M. Suvarna et al. explores text mining using transformer language models, an area of research that is certainly timely and carries immense potential for the future of data analysis. The authors build upon an earlier study (Vaucher, A. C. et al. Automated extraction of chemical synthesis actions from experimental procedures. Nat. Commun. 11, 3601 (2020)) showcasing how the new transformer-based model surpasses the preceding approach in performance.

The submitted manuscript also provides recommendations by the authors for formatting experimental articles, aimed at simplifying data extraction. These recommendations will improve data extraction in the future. However, I would recommend expanding the discussion to include considerations on the use of electronic lab notebooks and repositories for publishing data. As the authors pointed out, managing and storing data in a FAIR (findable, accessible, interoperable, reusable) manner becomes increasingly critical. For instance, the guidelines proposed in the article at <https://www.nature.com/articles/s41586-022-04501-x> could be taken into account.

There are also a few minor issues that should be addressed to polish this otherwise interesting study. In Figure 3, there appears to be an error: for 'centrifuge,' there is no 'ground truth,' and the 'ground truth' for 'transfer' seems to have a higher frequency than for 'wait.' These inconsistencies might confuse readers.

Finally, I encountered an issue when trying to access the authors' web application – an error message was displayed, preventing me from testing it. Given that the web application is likely an integral part of the authors' work and a means for readers to engage further with the research, it is crucial that this technical issue be addressed.

Overall, this article contributes valuable insights to the field of text mining. The authors have taken a significant step forward by applying transformer language models, and I anticipate that the article as well as the web application will be well received by both theoretical and experimental researchers.

Reviewer #2 (Remarks to the Author):

In this work, the authors developed a natural language processing (NLP) model based on a recent transformer architecture for the extraction of consistent chemical synthesis procedures from literature sources. The NLP model is trained to extract tokens representing synthesis unit operations and their associated metadata such as temperatures, amounts, times, and chemical species. The training data is initially sourced from single-atom catalyst literature but shows promise for being expanded to other fields given additional annotated training data and specialized tokens for field-specific synthetic operations. The authors offer suggestions for the standardization of synthetic procedure method sections in papers. The standardizations involve grouping supplier information, minimizing extraneous information, removing calculations in favor of direct quantities, and tabulating variations in procedure instead of listing multiple variations in a single step. Overall, this paper is very well written and brings up some valid points for structuring procedures for ease of data mining in chemical and materials sciences. However, due to the specialized target audience of the NLP model of this paper, I don't find it suitable for the broad audience of Nature Communications. This work might be more suitable for a specialized journal focused on NLP models in chemistry.

Other comments:

1. Was the effect of increasing the training set size on the model accuracy investigated?
2. I would have liked to see more model architecture and hyperparameter details in the SI for the readers to better understand the NLP model.
3. How would you reconcile different procedure standardization between extraction via NLP and other data mining techniques?
4. Is this model intended to expand to be generative or to contain synthesis results like yield and

performance of the synthesized material? If so would you expect the NLP model to have some of the same issues as e.x. ChatGPT, where it only looks for token strings that sound correct regardless of truth?

NCOMMS-23-18806-T- Response to Reviewers

Comments in *blue* - Replies in black - Actions in **bold**

Indicated page, line, or figure numbers refer to the revised manuscript and/or supplementary information with changes highlighted

Reviewer #1

The manuscript by M. Suvarna et al. explores text mining using transformer language models, an area of research that is certainly timely and carries immense potential for the future of data analysis. The authors build upon an earlier study (Vaucher, A. C. et al. Automated extraction of chemical synthesis actions from experimental procedures. Nat. Commun. 11, 3601 (2020)) showcasing how the new transformer-based model surpasses the preceding approach in performance. The submitted manuscript also provides recommendations by the authors for formatting experimental articles, aimed at simplifying data extraction. These recommendations will improve data extraction in the future. Overall, this article contributes valuable insights to the field of text mining. The authors have taken a significant step forward by applying transformer language models, and I anticipate that the article as well as the web application will be well received by both theoretical and experimental researchers

We appreciate Reviewer #1's positive feedback, recognizing the timeliness and relevance of our study to the broad audience of both experimental and theoretical researchers. Their valuable recommendations have only helped to improve the impact of our study, as detailed below.

1. I would recommend expanding the discussion to include considerations on the use of electronic lab notebooks and repositories for publishing data. As the authors pointed out, managing and storing data in a FAIR (findable, accessible, interoperable, reusable) manner becomes increasingly critical. For instance, the guidelines proposed in the article at <https://www.nature.com/articles/s41586-022-04501-x> could be taken into account?

We appreciate the Reviewer's suggestion and agree with the relevance. While it is not yet common practice in heterogeneous catalysis, the establishment of standardized methodologies for capturing synthetic protocols within electronic lab notebooks and subsequently uploading them to repositories as associated datasets would indeed greatly contribute to cultivating a FAIR compliant data ecosystem. Our transformer model primarily served to extract insights from published data until these practices become standardized. **We have extended these discussions in pages 17-18, lines 437-457.**

2. There are also a few minor issues that should be addressed to polish this otherwise interesting study. In Figure 3, there appears to be an error: for 'centrifuge,' there is no 'ground truth,' and the 'ground truth' for 'transfer' seems to have a higher frequency than for 'wait.' These inconsistencies might confuse readers.

We thank the Reviewer for pointing out these issues and have included the ground truth information of the action term 'centrifuge' in the revised Fig. 5a. As for the frequencies of the action terms 'transfer' and 'wait', we re-evaluated the data and found them to be quite comparable. However, we acknowledge that the model's prediction frequency for 'wait' is slightly lower than for 'transfer', which could stem from the model's limitations in capturing actions associated with 'wait' in certain instances. **We revised the original Fig. 3b with the necessary edits and present it as Fig. 5a in the revised draft.**

3. Finally, I encountered an issue when trying to access the authors' web application – an error message was displayed, preventing me from testing it. Given that the web application is likely an integral part of the authors' work and a means for readers to engage further with the research, it is crucial that this technical issue be addressed.

We thank the Reviewer for pointing out the technical issue with accessing our web application. We fully recognize the significance of our web application as a pivotal component for deepening the engagement of readers. **We took immediate action to address the software issue**, and confirm the web application is now fully operational for interested readers to explore and utilize.

Reviewer #2

In this work, the authors developed a natural language processing (NLP) model based on a recent transformer architecture for the extraction of consistent chemical synthesis procedures from literature sources. The NLP model is trained to extract tokens representing synthesis unit operations and their associated metadata such as temperatures, amounts, times, and chemical species. The training data is initially sourced from single-atom catalyst literature but shows promise for being expanded to other fields given additional annotated training data and specialized tokens for field-specific synthetic operations. The authors offer suggestions for the standardization of synthetic procedure method sections in papers. The standardizations involve grouping supplier information, minimizing extraneous information, removing calculations in favor of direct quantities, and tabulating variations in procedure instead of listing multiple variations in a single step. Overall, this paper is very well written and brings up some valid points for structuring procedures for ease of data mining in chemical and materials sciences. However, due to the specialized target audience of the NLP model of this paper, I don't find it suitable for the broad audience of Nature Communications. This work might be more suitable for a specialized journal focused on NLP models in chemistry.

We appreciate the Reviewer's input and for noting our contribution's valuable messages and clarity. Although our work entails developing a natural language processing model, its core objective was more extensive: exploring whether machine learning can expedite heterogeneous catalyst synthesis planning. This endeavor encompassed creating a customized state-of-the-art model, analyzing results, identifying protocol standardization gaps, and proposing a roadmap for improving machine readability. We are fully convinced that the broad scope of Nature Communications makes it the right platform to publish our contribution, as it bridges the still distant communities of experimentalists and data scientists in the field of chemistry, cross-fertilizing them. This aspect, which was recognized by Reviewer #1, can catalyze the proliferation of similar studies accelerating innovation. We have carefully revised the manuscript to better underscore these crucial aspects.

1. Was the effect of increasing the training set size on the model accuracy investigated?

We appreciate the Reviewer's inquiry. We have evaluated the effect on the model accuracy by training it on varying proportions (25%, 50%, 75%, and 100%) of the training split from the annotated data. We also assessed the impact of data augmentation by artificially increasing the size of each of the above split by a factor 10 by substituting durations, temperatures, compound names, quantities, etc. We devised eight scenarios to investigate the effect of increasing the training set size on the model accuracy. **We discuss these results in the Supplementary Information pages 13-14, Supplementary Note 4, and Supplementary Fig. 2.**

Based on these tests, we observe that model accuracies cannot be improved solely based on the data quantity. Instead, our critical insight is that the model accuracy strongly hinges on the data quality, *i.e.*, consistency and standardization in reporting synthesis protocols is paramount, as emphasized throughout the study. We provide further details in the manuscript **pages 16-17, lines 418-436 and in Fig 6c.**

2. I would have liked to see more model architecture and hyperparameter details in the SI for the readers to better understand the NLP model.?

We thank the Reviewer for this pertinent comment, as the transformer model is the very base of this study. **We have now presented a more detailed description of the model architecture and its**

hyperparameters in the Supplementary Information pages 10-12, Supplementary Note 3, and Supplementary Tables 5-6.

3.3. How would you reconcile different procedure standardization between extraction via NLP and other data mining techniques?

We appreciate the interesting question raised by the Reviewer. While we have not quantitatively compared the effects of different standardization approaches on various data mining tools, we emphasize that consistent protocol reporting benefits all data mining methods. Standardized data, whether structured or unstructured, enhance the reliability and uniformity of inputs and is required to fully capitalize on emerging text mining or machine learning-related tools, requiring a change in mindset in current reporting practices. **We discuss these points as merits of data standardization in pages 17-18, lines 437-456.**

4. Is this model intended to expand to be generative or to contain synthesis results like yield and performance of the synthesized material? If so would you expect the NLP model to have some of the same issues as e.x. ChatGPT, where it only looks for token strings that sound correct regardless of truth?

We thank the Reviewer for this intriguing question. Given the limitations in data quality we encountered, it's important to note that our priority lies in addressing the immediate need for standardization of the reporting of synthetic protocols. Nevertheless, a possible future direction involves making the model generative. To accomplish the same, we envision using GPT-4's prompt engineering and re-training our sAC transformEr (ACE) model on the modified paragraphs. The inclusion of performance data in the model is also feasible but will face similar issues until progress towards improving data reliability and consistency is achieved. **We have discussed the model limitations in pages 13-14, lines 330-344, and the future scope of the model in page 19, lines 499-502.**

REVIEWER COMMENTS

Reviewer #1 (Remarks to the Author):

In my opinion the authors addressed all reviewer questions. Based on this thorough revision, I believe the manuscript is now well-prepared for publication in Nature Communications.

Reviewer #2 (Remarks to the Author):

The authors have properly addressed the questions and concerns raised by the reviewer.

Reviewer #4 (Remarks to the Author):

The manuscript "Language models and protocol standardization guidelines for accelerating synthesis planning in heterogeneous catalysis" describes an NLP approach to extracting synthesis procedures from existing literature. At the example of single-atom catalysts, the authors report a transformer model for extracting pre-defined action items from plain-text descriptions of experiments, trained on a small manually annotated dataset of 127 procedures, and provide a web application for simplified usage of their model. Based on the findings when validating and applying the model, the authors propose a series of guidelines for improving the data quality in the heterogeneous catalysis literature.

The article addresses the highly topical issue of data reporting and data standardization, which is of high interest for computational and experimental scientists from various disciplines. Related efforts and views have been extensively discussed in the past years, especially in the field of organic synthesis (amongst others, by parts of the authors: Vaucher et al, Nat. Commun. 2020, 10.1038/s41467-020-17266-6), and the translation to solid-state materials is logical. Overall, the article is clearly structured and well-written. However, I have a number of questions and concerns that need to be addressed before I can recommend this paper for publication.

1. From a conceptual standpoint, the attempt to define a standardized "language" (with defined action items) strongly resembles the XDL work by Cronin and co-workers, who have attempted to do the same for organic synthesis (Science 2020, 10.1126/science.abc2986). In this work, they also define an NLP system (SynthReader) to convert natural language procedures to XDL. Recently, Skreta et al. (arXiv 2023, 10.48550/arXiv.2303.14100) showed significantly improved performance in this task using iterative GPT-3 prompting. At minimum, these works should be cited. How do the authors place their work in this context?

2. I am confused by the authors' guidelines for standardizing the experimental procedures: Whilst I fully agree with the necessity to report all (!) experimental procedures, along with the corresponding metadata (not discussed here), the recommendations contain a series of aspects that are specific to the limitations of their (and possibly other recent) NLP approaches (e.g. ii, iii, v, vi). For a true standardization of synthesis procedures, shouldn't the goal be to provide all procedures in a structured format, rather than in plain text with some formatting guidelines? Arguably, to enable human interoperability, these structured formats should be readily translatable to natural language (much easier than vice versa).

3. Ironically, while the authors advocate for FAIR data for enhanced reproducibility, I was not able to find the code required to reproduce the findings shown in the manuscript. Even though the authors provide a detailed description of the model architecture in the Supplementary Information (and the trained model hidden behind the web interface), is there any reason why the code to produce the findings in the paper itself is not available (as a Github repository or as a snapshot on Zenodo)? In my view, this should be mandatory in 2023, since it otherwise provides an unnecessary barrier to

reproducibility and interoperability.

Further remarks:

- The paragraph on the statistics of the extracted steps (l. 141 and following) mixes results from the analysis with expert knowledge in the field. In particular, the phrasing "The results of our analysis on the synthesis of SACs offer exciting insights" is misleading, since the majority of rationales and explanations in this paragraph are not provided by the analysis results, but by domain expertise.
- Along the same lines, l. 145: "Our analysis revealed that Fe is one of the most commonly investigated metals for the ORR reaction." This is well-established knowledge in the field, and not really a new insight from the analysis (in fact, it is shown in Figure 1c already)
- How is the train-validation-test splitting performed? How do the authors ensure that the results are not given for a "cherry-picked" test set?
- How is the extraction of parameters for action terms (e.g. material, quantity, ...) performed? How is the quality of parameter extraction validated?
- The segmentation of synthesis procedures into single sentences, and the evaluation of model accuracy on single sentences feels like a significant simplification to me. How is parsing of text paragraphs into sentences performed reliably? What happens if the information about a specific action item is spread over two sentences?
- The visualization of the model predictions in Fig. 5a raises a series of concerns: The diagram shows that the ACE model learns the distribution of action terms. Is this distribution different between and the train and the test set? If not, this figure does not contain any meaningful information about the model's predictive capabilities. A confusion matrix could be a way more informative visualization here, to a) better visualize the actual model predictivity, and b) learn about the model's failure modes.
- The definition of the HMI score raises some questions. What constitutes a "partially correct action"? In which sense does the HMI score require less human labor than manual annotation? How is the correct order of events scored?
- The authors repeatedly use the term "bias-free" (l. 111, l. 202) to describe their datasets. I advocate for a careful use of this term (since there is a range of underlying biases in the data), and would recommend the use of "unbiased selection" / "randomized subset" etc. instead.

NCOMMS-23-18806B - Response to Reviewers

Comments in *blue* - Replies in black - Actions in **bold**

Indicated page, line, or figure numbers refer to the revised manuscript and/or supplementary information with changes highlighted

Reviewer #1

In my opinion, the authors addressed all reviewer questions. Based on this thorough revision, I believe the manuscript is now well-prepared for publication in Nature Communications.

We thank the Reviewer for their time and consideration in reviewing our work and for recommending it for publication in Nature Communications.

Reviewer #2

The authors have properly addressed the questions and concerns raised by the reviewer.

We thank the Reviewer for appreciating our work's novelty and interdisciplinary nature and recommending it for publication in Nature Communications.

Reviewer #4

The manuscript “Language models and protocol standardization guidelines for accelerating synthesis planning in heterogeneous catalysis” describes an NLP approach to extracting synthesis procedures from existing literature. At the example of single-atom catalysts, the authors report a transformer model for extracting pre-defined action items from plain-text descriptions of experiments, trained on a small manually annotated dataset of 127 procedures, and provide a web application for simplified usage of their model. Based on the findings when validating and applying the model, the authors propose a series of guidelines for improving the data quality in the heterogeneous catalysis literature.

The article addresses the highly topical issue of data reporting and data standardization, which is of high interest for computational and experimental scientists from various disciplines. Related efforts and views have been extensively discussed in the past years, especially in the field of organic synthesis (amongst others, by parts of the authors: Vaucher et al, Nat. Commun. 2020, 10.1038/s41467-020-17266-6), and the translation to solid-state materials is logical. Overall, the article is clearly structured and well-written. However, I have a number of questions and concerns that need to be addressed before I can recommend this paper for publication.

We appreciate the positive feedback from the Reviewer, recognizing the relevance and quality of our study. Their constructive criticism and model recommendations helped us to further improve the impact of our study, as detailed below.

1. From a conceptual standpoint, the attempt to define a standardized “language” (with defined action items) strongly resembles the XDL work by Cronin and co-workers, who have attempted to do the same for organic synthesis (Science 2020, 10.1126/science.abc2986). In this work, they also define an NLP system (SynthReader) to convert natural language procedures to XDL. Recently, Skreta et al. (arXiv 2023, 10.48550/arXiv.2303.14100) showed significantly improved performance in this task using iterative GPT-3 prompting. At minimum, these works should be cited. How do the authors place their work in this context?

We thank the Reviewer for the valuable suggestions. The current study, which builds upon previous research (*Nat. Commun.* **11**, 3601 (2020)), extends its application to the domain of heterogeneous catalysis. It shares a common foundation with the XDL work by Cronin and co-workers (*Science* **370**, 101-108 (2020)), as both are based on the use of structured frameworks for defining specific sets of actions and their associated properties to describe laboratory operations.

The SynthReader’s NLP system falls in the category of algorithms not relying on deep learning. A conversion between the two NLPs for organic synthesis may be feasible, although we are not aware this has been implemented anywhere yet, and the exact extraction pipeline of the SynthReader is not described in detail in the paper. On the other hand, the article of Skreta et al. reports a deep-learning-based extraction of synthesis instructions. The approaches in both studies have notable distinctions to our current model in that i) their extraction produces instructions in XDL format, ii) the models are fine-tuned starting from a

more general pretrained model (GPT-3), and iii) are tasked with self-correction in an iterative manner.

We have now cited these references in the introduction on page 2, lines 20-22, along with other relevant works related to the extraction of synthesis steps from text. We did not extend the description of other text mining approaches or language models, as the primary focus of our study relates to their application in the field of heterogeneous catalysis. An extensive discussion including all related works is beyond the scope of the current work.

2. I am confused by the authors' guidelines for standardizing the experimental procedures: Whilst I fully agree with the necessity to report all (!) experimental procedures, along with the corresponding metadata (not discussed here), the recommendations contain a series of aspects that are specific to the limitations of their (and possibly other recent) NLP approaches (e.g. ii, iii, v, vi). For a true standardization of synthesis procedures, shouldn't the goal be to provide all procedures in a structured format, rather than in plain text with some formatting guidelines? Arguably, to enable human interoperability, these structured formats should be readily translatable to natural language (much easier than vice versa).

Reviewer #4 raises an important issue: the need for standardized experimental procedures. We share the Reviewer's perspective on the significance of structured formats for synthesis protocols, which can significantly enhance data uniformity and facilitate the application of machine learning techniques to derive collective insights in chemistry and catalysis. Our guideline # vi specifically addresses the presentation of protocols in tabular formats, underlining the alignment of our vision. However, we acknowledge that the practical implementation of structured protocols in the near future remains an open question, requiring collaborative initiatives within the research community. An alternative approach could involve presenting synthesis procedures as sequences of actions, akin to the output of our model rather than in lengthy prose. We believe such practices would not only facilitate the use of existing text mining tools but also improve the reproducibility of experiments, which presents a long-standing challenge. **We discuss these aspects on pages 12 and 13, lines 19-28 and 1-6, respectively.**

Regarding the Reviewer's concern that our guidelines *ii* (list catalyst and synthesis method exclusively), *iii* (provide supplier details in a uniform manner), *v* (concise writing but all key details), and *vi* (tabular format for similar procedures) could be specific limitations of NLP approaches, in our viewpoint *ii*, *v*, and *vi* are general good practice for writing synthetic protocols to ensure that all relevant details are provided. Through such data standardization efforts, which inadvertently improve the quality of data, we anticipate significant benefits to the general class of text mining approaches, be it natural language processing, association rule learning, or sequential pattern mining, etc. The implementation of point *iii* is arguably specific to our approach. However, we encourage the provision of supplier details in a uniform manner. **We have modified Figure 6 to reflect that and have discussed the best approach for our model in the revised manuscript.**

3. Ironically, while the authors advocate for FAIR data for enhanced reproducibility, I was not able to find the code required to reproduce the findings shown in the manuscript. Even though the authors provide a detailed description of the model architecture in the Supplementary Information (and the trained model hidden behind the web interface), is there any reason why the code to produce the findings in the paper itself is not available (as a GitHub repository or as a snapshot on Zenodo)? In my view, this should be mandatory in 2023, since it otherwise provides an unnecessary barrier to reproducibility and interoperability.

We thank the Reviewer for highlighting this point. Much of the training process relied on the “paragraph2actions” repository (<https://github.com/rxn4chemistry/paragraph2actions>), already accessible on GitHub. In addition to the web interface and its code that were already available. **We have now open-sourced the remaining components for the current work and uploaded it to GitHub: <https://github.com/rxn4chemistry/sac-action-extraction>.** In the current version of the draft, **we also added a “Code Availability” section on page 20, lines 1-4** directing interested readers to the GitHub link, with all necessary codes required to replicate and build upon our study’s results.

Further remarks:

4. The paragraph on the statistics of the extracted steps (l. 141 and following) mixes results from the analysis with expert knowledge in the field. In particular, the phrasing “The results of our analysis on the synthesis of SACs offer exciting insights” is misleading, since the majority of rationales and explanations in this paragraph are not provided by the analysis results, but by domain expertise. Along the same lines, l. 145: “Our analysis revealed that Fe is one of the most commonly investigated metals for the ORR reaction.” This is well-established knowledge in the field, and not really a new insight from the analysis (in fact, it is shown in Figure 1c already)

We appreciate the Reviewer’s important feedback. Indeed, the rationale and explanations of the model findings draw heavily on domain knowledge. It’s important to note that our model’s analysis aligns with existing literature, providing a validation of its predictive capabilities. To reflect this and ensure clarity, **we have rephrased the above-mentioned sentences accordingly in page 6, lines 4-5.**

5. How is the train-validation-test splitting performed? How do the authors ensure that the results are not given for a “cherry-picked” test set?

The train-validation-test split was carried out by a randomized process at the sentence level. First, the number of sentences assigned to the train, validation, and test sets were determined based on the desired sizes (as a percentage of the total). Second, the full set of sentences and associated annotations are shuffled, randomizing their order. Third, the shuffled dataset is split into three contiguous subsets of the desired sizes for the respective splits. **We have included these details in the Methods section, page 16, lines 12-19.**

For those interested, the code for this splitting process is now available on GitHub:

https://github.com/rxn4chemistry/sac-actionextraction/blob/main/src/sac_action_extraction/create_annotation_splits.py

It relies on functionality in the paragraph2actions repository:

https://github.com/rxn4chemistry/paragraph2actions/blob/main/src/paragraph2actions/data_splitting.py

This approach ensures that the results are based on randomized selection, mitigating the risk of “cherry-picking” and providing a more robust evaluation of the model’s performance.

6. How is the extraction of parameters for action terms (e.g. material, quantity, ...) performed? How is the quality of parameter extraction validated?

After the extraction of actions using the translation model, the results are converted to Python objects with associated properties. **We have developed user-defined functions for extracting various parameters such as compound names, quantities, or temperatures and presented these details in the Supplementary Information on Page 9, lines 19-24.** You can explore a detailed **example of this process in our GitHub repository:**

<https://github.com/rxn4chemistry/sac-action-extraction#analysis-of-extracted-actions>.

To assess the quality of the parameter extracted, we introduced the Human Machine-readability Index (HMI). This index exploits domain knowledge from heterogeneous catalysis experts to quantify and validate the model’s parameter predictions. **Further details are provided in response to comment #9 of the Reviewer, which is related.**

7. The segmentation of synthesis procedures into single sentences, and the evaluation of model accuracy on single sentences feels like a significant simplification to me. How is parsing of text paragraphs into sentences performed reliably? What happens if the information about a specific action item is spread over two sentences?

Paragraph-to-sentence segmentation is handled modularly, and any subclass of SentenceSplitter from our repository can be used:

https://github.com/rxn4chemistry/paragraph2actions/blob/main/src/paragraph2actions/sentence_splitting/sentence_splitter.py

In the example mentioned, we employ an implementation based on ChemDataExtractor: <https://github.com/rxn4chemistry/sac-action-extraction#analysis-of-extracted-actions>

Regarding information distributed over multiple sentences, our approach is based on practical observations from previous work, which found that experimental procedures usually contain few cross-sentence dependencies (*Nat. Commun.* **11**, 3601 (2020)). That work also showed that, in most cases, information contained in consecutive sentences could also be determined from context when analyzing the action terms. However, we agree that some cross-dependencies over multiple sentences do occur. **We extended these discussions in the Supplementary Information on Page 9, lines 10-18.** In scenarios with more data or larger models, direct extraction of actions from whole paragraphs could be a viable alternative.

8. The visualization of the model predictions in Fig. 5a raises a series of concerns: The diagram shows that the ACE model learns the distribution of action terms. Is this distribution different between and the train and the test set? If not, this figure does not contain any meaningful information about the model's predictive capabilities. A confusion matrix could be a way more informative visualization here, to a) better visualize the actual model predictivity, and b) learn about the model's failure modes.

The comparison of the distribution of predicted versus ground truth actions presented in Fig. 5a is crucial for understanding the ACE model's predictive capabilities in the context of action extraction. Specifically, the visualization allows us to identify actions where the performance is suboptimal, which guides us in refining the model. While we agree that a confusion matrix might be a valuable tool for specific classification tasks (typically when a model predicts one category instead of another), the open-ended nature of action extraction from heterogeneous catalyst synthesis protocols makes a distribution analysis more appropriate in this context. We note that the data in Fig. 5 correspond to the test split, which enables meaningful comparison. To clarify this point, **we have modified the caption of Fig 5a.**

9. The definition of the HMI score raises some questions. What constitutes a "partially correct action"? In which sense does the HMI score require less human labor than manual annotation? How is the correct order of events scored?

We appreciate the Reviewer's query. Since our model converts paragraphs into action sequences, the HMI (human machine-readability index) score definition is as follows:

- If the model correctly predicts action terms and extracts corresponding material properties or unit operation values, it receives a total score of 1.
- If the model predicts the correct action term but extracts incorrect material properties or unit operation values, then it gets a score of 0.5, applied for partially correct actions.
- If the model incorrectly predicts an action term, it receives a score of 0.

By design, the HMI score incorporates a human factor in evaluating our model's performance since annotated ground truth data may not be available. This offers a practical approach for heterogeneous catalysis researchers, our end users, to compare the performance based on a small subset of paragraphs, while also exploiting their domain knowledge to quantify and validate the model's parameter predictions. Calculating HMI scores requires much less effort than large-scale projects, and is more accessible to those without advanced coding skills. **We have extended the discussion on HMI scoring on page 17, lines 11-15 and page 18, lines 1-6.**

10. The authors repeatedly use the term “bias-free” (l. 111, l. 202) to describe their datasets. I advocate for a careful use of this term (since there is a range of underlying biases in the data), and would recommend the use of “unbiased selection” / “randomized subset” etc. instead.

We thank the Reviewer for the recommendation. **We have replaced the term bias-free with randomized subset throughout the manuscript on page 5, line 1 and page 8, line 10.**

REVIEWERS' COMMENTS

Reviewer #4 (Remarks to the Author):

The authors have provided a revised version of their manuscript, which convincingly addresses of my concerns, particularly in terms of accessibility and reproducibility (code availability, detailed descriptions of methods). I am therefore happy to recommend the paper for publication in Nature Communications, and want to congratulate the authors on an important piece of work, extending the ideas of procedure digitization and standardization to heterogeneous catalyst synthesis.